# UGLAD: A DEEP LEARNING MODEL TO RECOVER CONDITIONAL INDEPENDENCE GRAPHS

## ABSTRACT

Probabilistic Graphical Models are generative models of complex systems. They rely on conditional independence assumptions between variables to learn sparse representations which can be visualized in a form of a graph. Such models are used for domain exploration and structure discovery in poorly understood domains. This work introduces a novel technique to perform sparse graph recovery by optimizing deep unrolled networks. Assuming that the input data $X \in \mathbb{R}^{M \times D}$ comes from an underlying multivariate Gaussian distribution, we apply a deep model on $X$ that outputs the precision matrix $\Theta$. Then, the partial correlation matrix P is calculated which can also be interpreted as providing a list of conditional independence assertions holding in the input distribution. Our model, uGLAD[1], builds upon and extends the state-of-the-art model GLAD Shrivastava et al. (2020b) to the unsupervised setting. The key benefits of our model are (1) uGLAD automatically optimizes sparsity-related regularization parameters leading to better performance than existing algorithms. (2) We introduce multi-task learning based 'consensus' strategy for robust handling of missing data in an unsupervised setting. We evaluate performance on synthetic Gaussian, non-Gaussian data generated from Gene Regulatory Networks, and present case studies in anaerobic digestion and infant mortality.

*Keywords*: Graphical Lasso, Deep Learning, Unrolled Algorithms, Conditional Independence graphs, Sparse graphs

## 1 INTRODUCTION

Probabilistic graphical models (PGMs) Pearl (1988); Koller & Friedman (2009) are generative models of complex systems, used to describe dependencies within a set of random variables and visualize the structure of a domain. The models rely on conditional independence assumptions between variables, which result in sparse representation and enable efficient inference. In the graphical representation of the models, conditional independence is indicated by an absence of an edge between two variables. Such models can be learned from observational data Heckerman (1995); Friedman et al. (2008). Structure discovery enabled by PGMs is important for new and poorly understood domains where relationships between variables are not known. PGMs have been used in various domains, including medical diagnosis Heckerman et al. (1992); Heckerman & Nathwani (1992), fault diagnosis, genomic data analysis via gene regulatory networks Moerman et al. (2019); Pratapa et al. (2020); Aluru et al. (2021); Shrivastava et al. (2022), speech recognition, and in finance Hallac et al. (2017).

The problem of recovering the structure from observational data is particularly difficult in high-dimensional settings, where the number of features may be larger than the number of observations. We focus specifically on learning undirected models where the data is assumed to have been generated from a multivariate Gaussian distribution Friedman et al. (2008); Belilovsky et al. (2017); Zheng et al. (2018); Yu et al. (2019); Shrivastava et al. (2020b). In such cases, the goal is to estimate a sparse inverse covariance matrix. Sparsity is typically enforced by the use of $\ell_1$ (lasso) regularization.

Assume we have a $d$-dimensional multivariate Gaussian random variable $X = [X_1, \dots, X_d]^\top$ with $m$ observations. The goal is to estimate its covariance matrix $\Sigma^*$ and precision matrix $\Theta^* = (\Sigma^*)^{-1}$. $\Theta^*$ encodes conditional independence assumptions between variables: the $ij$-th component is zero

---

[1] uGLAD code link (Provided in Supplementary)

if and only if $X_i$ and $X_j$ are conditionally independent given the other variables $\{X_k\}_{k \neq i,j}$. This problem is known as the sparse graph recovery problem and usually formulated (following Friedman et al. (2008)), as the $\ell_1$-regularized maximum likelihood estimation

$$\widehat{\Theta} = \arg\min_{\Theta \in \mathcal{S}_{++}^d} - \log(\det \Theta) + \operatorname{tr}(\widehat{\Sigma}\Theta) + \rho \left\| \Theta \right\|_{1,\text{off}}, \tag{1}$$

where $\widehat{\Sigma}$ is the empirical covariance matrix based on $m$ samples, $\mathcal{S}_{++}^d$ is the space of $d \times d$ symmetric positive definite matrices (SPD), and $\left\| \Theta \right\|_{1,\text{off}} = \sum_{i \neq j} |\Theta_{ij}|$ is the off-diagonal $\ell_1$ regularizer with regularization parameter $\rho$. The use of this estimator is justified even for non-Gaussian $X$, since it is minimizing an $\ell_1$-penalized log-determinant Bregman divergence Ravikumar et al. (2011). The problem in Eq. 1 is a convex optimization problem. It can be solved by many algorithms, see Section 2 for examples.

However, classic approaches have their limitations in both the statistical aspect and computational aspect. Statistically, the classic formulation uses a single regularization parameter $\rho$ for all entries in the precision matrix $\Theta$, which may not be optimal. A recent theoretical work Sun et al. (2018) validates the use of adaptive parameters. Shrivastava et al. (2020b) has proposed a model with multiple regularization parameters called GLAD and has shown emprical evidence that such a model pushes the sample complexity limits. Based on that evidence, we hypothesise that one may obtain better recovery results by allowing the regularization parameters to vary across the entries in the precision matrix. However, it is hard for traditional approaches to search over a large number of hyperparameters. Computationally, the complexity of solving the optimization depends on the convexity of the objective, the step sizes of the algorithm, the initialization, the design of the update steps, etc. Different problems may require different designs of the algorithm to achieve a better efficiency. A unified design for all problems may not be optimal.

In this work, we propose uGLAD (unsupervised-GLAD), which is a deep model that can recover sparse graphs in an unsupervised manner. As the name suggests, it builds upon and extends the GLAD model, which recovers sparse graphs under supervision. uGLAD uses the same objective function as GLAD . Using an additional variable $Z$, the $\ell_1$-regularized maximum likelihood from Eq. 1 can be written as

$$\widehat{\Theta} = \arg\min_{\Theta \in \mathcal{S}_{++}^d} - \log(\det \Theta) + \operatorname{tr}(\widehat{\Sigma}\Theta) + \rho \left\| Z \right\|_1, s.t. \quad Z = \Theta$$

Now, including the constraint as squared penalty term $\lambda$ we obtain the reformulated objective as

$$\widehat{\Theta}_\lambda, \widehat{Z}_\lambda = \arg\min_{\Theta, Z \in \mathcal{S}_{++}^d} - \log(\det \Theta) + \operatorname{tr}(\widehat{\Sigma}\Theta) + \rho \left\| Z \right\|_1 + \tfrac{1}{2}\lambda \left\| Z - \Theta \right\|_F^2 \tag{2}$$

Note that introducing the variable $Z$ helps in splitting the objective into 2 parts and those can be optimized alternately using the Alternating Minimization algorithm. A Conditional Independence (CI) graph can be obtained from this precision matrix by calculating the partial correlation matrix. The CI graphs are very informative as they describe how the features in a domain interact with each other. Furthermore, they can additionally capture negative partial correlations between features which are usually not modeled by traditional graphs.

Key contributions of this work:

- *Extending GLAD to unsupervised setting*: The uGLAD doesn't rely on availability of ground truth to do graph recovery.
- *Adaptive hyperparameters*: The uGLAD architecture design enables the hyperparameters to optimally adapt at each step of the unrolled Alternating Minimization (AM) algorithm Shrivastava et al. (2020b) that leads to its superior performance.
- *Automatically decide optimum sparsity parameters*: The sparsity of the recovered graph is highly sensitive to the choice of the regularization hyperparameters. Instead, uGLAD models hyperparameters within the neural network framework and they are directly optimized for the uGLAD objective defined above. So, there is no need to separately optimize the sparsity hyperparameters which is otherwise a computationally expensive process.
- *Runtime efficiency*: The uGLAD software can run on GPUs for higher time efficiency and scalability.
- *Missing data handling*: The uGLAD framework can also be used for multi-task learning. We leverage this property further to develop a novel 'consensus' strategy to robustly handle missing data.

## 2 RELATED WORK

**Traditional algorithms for graphical lasso**: These are primarily iterative methods for optimizing the graphical lasso objective. Main methods developed for this problem in the last two decades are detailed in this survey paper Witten et al. (2011). The variant of the Block Coordinate Descent (BCD) algorithm by Friedman et al. (2008) is widely used to recover graphs using the graphical lasso method. This method is also implemented in the popular python `sklearn` package. The G-ISTA algorithm by Guillot et al. (2012) is based on the iterative shrinkage thresholding procedure. It is one of the prominent methods based on using the proximal gradient descent approach. The Alternating Direction Method of Multipliers (ADMM) Danaher et al. (2014) has also been successfully used in various graphical lasso based applications.

**Deep Learning approaches for graph recovery**: DeepGraph (DG) Belilovsky et al. (2017), is a supervised deep learning method that takes in the input samples and outputs the corresponding adjacency matrix which shows the connections between input features. DeepGraph architecture consists of many convolutional layers followed by multi-layer perceptrons that finally decides whether an edge is present between every combination of features. Another relevant DL method roughly based on modeling the input data with a Variational Autoencoder for graph recovery is DAG-GNN Yu et al. (2019). These deep architectures have very high number of learnable parameters, which is a significant drawback. Hence, we pursue a different line of research (using inductive biases) which gives similar performance with significantly reduced number of learnable parameters and brings more interpretability as shown in Shrivastava et al. (2020b).

**Deep learning models using inductive biases**: Improved performance often results from including domain knowledge in the design/initialization of deep learning architectures. For instance, Shrivastava et al. (2019) presents a generic technique to use a probabilistic graphical model as a prior to design a deep model. The authors were able to show enhanced performance on the document classification task by leveraging the Latent Dirichlet Allocation prior. Another way of including prior knowledge about the domain is using an optimization algorithm for a related objective function as a template to design the deep architecture. Unrolling the optimization algorithms and parameterizing the step updates using neural networks have been fairly successful for many tasks Liu & Chen (2019); Chen et al. (2020); Shrivastava (2020); Pu et al. (2021); Shrivastava et al. (2022).

This work focuses on recovering undirected graphs, specifically based on optimizing the graphical lasso objective function. Hence, we skipped discussing the methods developed specifically to recover Directed Acyclic Graphs (DAGs). The work most closely related to ours is the `GLAD` Shrivastava et al. (2020b) model. Since our algorithm builds upon `GLAD`'s architecture we are going to describe it in detail in Section 3, while pointing out ways in which `uGLAD` differs from `GLAD`.

## 3 THE uGLAD MODEL

Given input data $X \in \mathbb{R}^{M \times D}$, with $M$ samples with $D$ features, the task is to recover a sparse graph showing partial correlations between the $D$ features. Recovering the sparse graph (the adjacency matrix) corresponds to obtaining the precision matrix $\Theta$ and the partial correlation matrix P of the underlying multivariate Gaussian distribution.

### 3.1 UNDERSTANDING THE GLAD ARCHITECTURE

uGLAD uses the same architecture as `GLAD` . We we have a function $\Theta = f_{nn}(X)$, implemented as the `GLAD` architecture Shrivastava et al. (2020b). The `GLAD` model uses the Alternating Minimization algorithm updates, unrolled to some iterations, for the maximum likelihood objective as a template for its deep architecture design. Penalty constants $(\rho, \lambda)$ are replaced by problem dependent neural networks, $\rho_{nn}$ and $\Lambda_{nn}$. These neural networks are minimalist in terms of the number of parameters as the input dimensions are mere $\{3, 2\}$ for $\{\rho_{nn}, \Lambda_{nn}\}$ and outputs a single value. Appendix A summarizes the update equations for the unrolled AM based model, `GLAD`. This unrolled algorithm with neural network augmentation can be viewed as a highly structured recurrent architecture, see Fig. 1.

Note that the `GLAD` model has a low number of learnable parameters and maintains the permutation invariance w.r.t. the input. The design prior of the AM algorithm enforces the positive definite

constraint of the precision matrix at every step of its optimization. One can see the state of the recovered graph at each unrolled step of the AM optimization, thus giving insights about the learning process. The AM algorithm converges linearly for the graphical lasso objective. The architecture is described in more detail in in Appendix A and in the original `GLAD` paper Shrivastava et al. (2020b).

### 3.2 THE `uGLAD` LOSS FUNCTION

To learn the parameters in `GLAD` architecture, the authors used supervision in form of the true underlying graphs. They leveraged the interpretable nature of the `GLAD`'s deep architecture to define the loss for training. Specifically, each iteration of the model will output a valid precision matrix estimation and this allowed them to add auxiliary losses to regularize the intermediate results of `GLAD`, guiding it to learn parameters which can generate a smooth solution trajectory.

The authors used Frobenius norm in their loss function:

$$\mathcal{L}_{\text{GLAD}} := \frac{1}{n} \sum_{i=1}^{n} \sum_{k=1}^{K} \gamma^{K-k} \left\| \Theta_k^{(i)} - \Theta^* \right\|_F^2, \tag{3}$$

where $(\Theta_k^{(i)}, Z_k^{(i)}, \lambda_k^{(i)}) = \texttt{GLADcell}_f(\widehat{\Sigma}^{(i)}, \Theta_{k-1}^{(i)}, Z_{k-1}^{(i)}, \lambda_{k-1}^{(i)})$ is the output of the recurrent unit `GLADcell` at $k$-th iteration, $K$ is number of unrolled iterations, $\gamma \leq 1$ is a discounting factor and $\Theta^*$ is the ground truth precision matrix. Then the stochastic gradient descent algorithm was used to train the parameters $f$ in the `GLADcell`.

In contrast to `GLAD`, `uGLAD` is designed to work without supervision, so it cannot take advantage of the underlying true generating distribution in the form of the true covariance matrix. Thus, we need to replace `GLAD`'s MSE loss from Equation 3 with a new loss. Given a function $\Theta = f_{nn}(X)$, we optimize the log likelihood function given by

$$\mathcal{L}_{\text{uGLAD}}(S, \Theta) = -\log|\Theta| + \langle S, \Theta \rangle \tag{4}$$

$$\mathcal{L}_{\text{uGLAD}}(X) = -\log|f_{nn}(X)| + \langle \text{cov}(X), f_{nn}(X) \rangle \tag{5}$$

where $S = \text{cov}(X)$ is the covariance matrix. We take the function $f_{nn}$ as the `GLAD` model and substitute it in the `uGLAD` loss function.

Algorithm 1 gives the pseudo code for learning the `uGLAD` model for doing sparse graph recovery. Appendix A contains more information about the parameter choices of the `uGLAD` and the `GLAD` model.

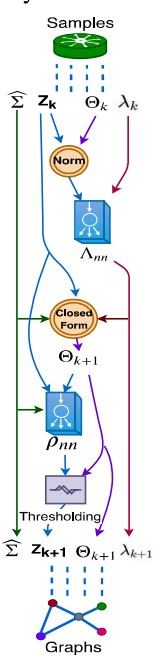

Figure 1: The recurrent unit `GLADcell`. (Taken from Shrivastava et al. (2020b), more details in Appendix A)

### 3.3 CONVERGENCE PROPERTIES OF `GLAD` AND `uGLAD`

The `GLAD` paper Shrivastava et al. (2020b) evaluates convergence properties for the `GLAD` algorithm using normalized mean square error (NMSE) and probability of success (PS) to evaluate the algorithm performance. NMSE is $\log_{10}(\mathbb{E} \| \Theta^p - \Theta^* \|_F^2 / \mathbb{E} \| \Theta^* \|_F^2)$ and PS is the probability of correct signed edge-set recovery, i.e., $\mathbb{P}\left[\text{sign}(\Theta_{ij}^p) = \text{sign}(\Theta_{ij}^*), \forall (i, j) \in \mathbf{E}(\Theta^*)\right]$, where $\mathbf{E}(\Theta^*)$ is the true edge set. Optimization objective always converges. However, errors of recovering true precision matrices measured by NMSE have very different behaviors given different regularity parameter $\rho$, which indicates the necessity of directly optimizing NMSE and hyperparameter tuning. NMSE values are very sensitive to both $\rho$ and the quadratic penalty $\lambda$ of ADMM method. In `GLAD` and `uGLAD`, $\rho$ and $\lambda$ are not fixed, but are optimized together with the rest of network parameters, leading to smooth convergence.

In experiments evaluating edge recovery success, `GLAD` consistently outperforms traditional methods in terms of sample complexity as it recovers the true edges with considerably fewer number of samples. Since in `uGLAD` we are still using the AM minimization based `GLAD` architecture which is also based on optimizing the Eq. 1, we expect the linear convergence properties of the AM algorithm will hold for `uGLAD` as well. The synthetic experiments in Sec. 6 show the results obtained from `uGLAD` are better or even surpass in comparison to block coordinate descent based approach.

### 3.4 Obtaining Conditional Independence graphs

The CI graph shows the partial correlations between the input features. A non-zero partial correlation between 2 features $(f_A, f_B)$ indicates a direct dependence. So, if all the other features are fixed, a positive partial correlation will indicate that increasing value of $f_A$ will increase the value of $f_B$ and vice-versa for the negative correlation. To obtain this CI graph, we calculate the partial correlation matrix from the precision matrix. Each entry of the partial correlation matrix $P_{ij}$ shows the correlation of the feature $x_i, x_j$ given the values of the other features are observed. This helps us obtain the direct dependence of the features, $P_{ij} = -\frac{\Theta_{ij}}{\sqrt{\Theta_{ii}\Theta_{jj}}}$. We can then visualize the graphs and study the positive & negative correlations, with edge weights corresponding to correlation strengths.

## 4 Multi-task learning for precision matrix recovery

Most of the work in learning Gaussian graphical models has focused on estimating a single model. In recent years, the framework was extended to jointly fitting a collection of such models, based on data that share the same variables, with dependency structure varying with some external category. For example, in NLP application, we can encounter different styles, which induce links between some concepts, even as the underlying grammar and semantics of the language stay the same.

---

**Algorithm 1:** Optimizing `uGLAD`

---

Input: Observations $\boldsymbol{X} \in \mathbb{R}^{M \times D}$
$S = \text{cov}(\boldsymbol{X})$;
**for** $e = 1, \cdots, \text{E}$ **do**
    $\hat{\Theta}_e = \text{GLAD}(S)$ unrolled for L iterations;
    Compute loss $\mathcal{L}_{\text{uGLAD}}(S, \hat{\Theta}_e)$;
    Backprop to update `GLAD` params;
**end for**
return $\hat{\Theta}_E$

---

There have been extensive studies on the joint estimation of multiple undirected Gaussian graphical models Song et al. (2009); Honorio & Samaras (2010); Guo et al. (2011); Cai et al. (2011); Oyen & Lane (2012); Danaher et al. (2014); Kolar et al. (2010); Mohan et al. (2014); Peterson et al. (2015); Yang et al. (2015); Gonçalves et al. (2016); Varici et al. (2021). Most traditional algorithms construct a joint objective for multiple estimation tasks. This objective typically incorporates similarities among various tasks by adding group norms or other regularizations. However, in many practical problems, we only know that multiple tasks are related, without knowing how they are similar to each other quantitatively. Manually constructing the joint objective may not best reflect the actual similarity.

In contrast to traditional algorithms, in `uGLAD`, we do not need to pre-assume the specific similarity among different tasks. Instead, we use a single network `uGLAD` to solve multiple tasks. Since the parameters in `uGLAD` are shared across different tasks, the similarity among the tasks is automatically learned from data. More specifically, given samples from $K$ different models, $\mathbf{X_K} = [X_1, X_2, \cdots, X_K]$, we optimize the following objective

$$\mathcal{L}_{\text{uGLAD-multitask}}(\mathbf{X_K}) = \frac{1}{K} \sum_{k=1}^{K} \mathcal{L}_{\text{uGLAD}}(\text{cov}(X_k), f_{nn}(X_k)) \tag{6}$$

Alternatively, we can use the cross-validation split $\{\mathbf{X_K}^{train}, \mathbf{X_K}^{val} \leftarrow \mathbf{X_K}\}$ for training.

## 5 Handling missing values

The missing data problem is ubiquitous in all data problems. `uGLAD` can easily be extended to handle this problem. If we observe that some specific feature columns have missing values and we have reasons to believe that these values are missing at random, we can run imputation algorithms for those columns to predict the missing entries. Then, we will have the complete imputed input data $X_{imp} \in \mathbb{R}^{M \times D}$ over which we can run the `uGLAD` model and obtain the underlying precision matrix.

It can often happen that there are technical errors or human mistakes in collecting samples, which can often lead to missing values or noise seeping into the sample. Also, we assume that the samples are independent and identically distributed (IID), so we cannot make use of the imputation techniques discussed in the case above. For this case, we propose a novel multi-task learning technique based on utilizing the `uGLAD`'s ability to optimize over a batch of input samples.

| AUPR | | | |
|---|---|---|---|
| Method | M=10 | M=25 | M=50 |
| BCD | 0.112±0.013 | 0.132±0.012 | 0.219±0.085 |
| uGLAD | 0.159±0.029 | 0.174±0.018 | 0.223±0.062 |
| AUC | | | |
| Method | M=10 | M=25 | M=50 |
| BCD | 0.505±0.007 | 0.532±0.031 | 0.617±0.083 |
| uGLAD | 0.572±0.027 | 0.595±0.044 | 0.651±0.021 |

Table 1: **Synthetic data: Gaussian**. AUPR and AUC on 20 test graphs for number of features $D = 25$ and varying number of samples $M$. Gaussian random graphs with sparsity $p = 0.1$ were chosen and edge values were sampled from $\sim \mathcal{U}(-1, 1)$. We can observe that uGLAD (CV mode) significantly outperforms the BCD algorithm (sklearn's graphicalLassoCV) for samples/features ratio « 1 and gives comparable performance as the number of samples increases.

**Consensus strategy: Multi-task learning over row-subsampled input**

The key idea is to create a batch of row subsamples of the input data $X \in \mathbb{R}^{M \times D}$. Since all of these subsamples come from the same underlying distribution, we should ideally recover the same precision matrix for the entire batch. Thus, if we have a model that can be jointly optimized over the entire batch for the uGLAD objective, resulting in the recovered precision matrix being robust against erroneous or noisy samples.

Steps for the multi-task learning approach to train the uGLAD model for handling missing data:

1. *Statistical imputation for the input*: Replace all the missing entries of the input data $X$ with their respective column mean (the mean is calculated ignoring the missing entries) $X[i, c] = mean(X[:, c])$. Replacing by mean is usually a preferred approach as its contribution zeros out $(X - \mu_X)$ while centering the data for the covariance matrix calculation.
2. *Getting the batches*: Perform stratified K-fold sampling to distribute the rows with missing values evenly among different batches. Say, we have $\mathbf{X_K} = [X_1, X_2, \cdots, X_K]$ batches with each $X_k \in \mathbb{R}^{M*(\frac{K-1}{K}) \times D}$. Thus, the batch input for the uGLAD model is $\mathbf{X_K} \in \mathbb{R}^{K \times M*(\frac{K-1}{K}) \times D}$.
3. *Optimizing uGLAD*: It becomes a multi-task learning setting as we are jointly optimizing over a batch input $\mathbf{X_K}$. The uGLAD model takes in the batch input $\mathbf{X_K}$ and outputs the corresponding $K$ precision matrices. Since, the entire batch of data is coming from the same underlying distribution, we use the entire data $X \in \mathbb{R}^{M \times D}$ for the uGLAD loss to optimize the parameters of the uGLAD model. Mathematically, we are minimizing the uGLAD loss over the batch as

$$\mathcal{L}_{\text{uGLAD-meta}}(\mathbf{X_K}) = \frac{1}{K} \sum_{k=1}^{K} \mathcal{L}_{\text{uGLAD}}(\text{cov}(X), f_{nn}(X_k)) \tag{7}$$

4. *Consensus among the batch to obtain the final precision matrix*: After optimizing the uGLAD for the batch input, we will obtain $K$ different precision matrices $\Theta_K \in \mathbb{R}^{K \times D \times D}$. Ideally, all the precision matrices should be the same but there will be some discrepancies as we are working with missing values. Our 'consensus' strategy to obtain the final precision matrix $\Theta^f$ is to find the common edges with their correlation type (positive or negative) from the batch precision matrices. Mathematically, we can obtain each entry $[i, j]$ of the final precision matrix as

$$\Theta_{i,j}^f = \text{max-count}_{k=1,...,K}(\text{sign } \Theta_{i,j}^k) \min_{k=1,...,K} |\Theta_{i,j}^k| \tag{8}$$

Here, the 'max-count' term determines whether the correlation among the batches for that entry is positive or negative. The $2^{nd}$ term chooses the minimum absolute value for that entry among the batches as this facilitates sparsity and is conservative in terms of the strength of an edge.

# 6 EXPERIMENTS

Our software package is hosted on the GitHub website, details given in Appendix 7. Its function signature is very much akin to the sklearn's GraphicalLassoCV Pedregosa et al. (2011). This was intended to make it easier for the users to try out our method with minimal change to their existing code pipeline. Appendix D lists some more potential applications of the uGLAD model. We believe that our model can be seamlessly integrated with the existing pipeline for these applications and we

| AUPR | | | |
|---|---|---|---|
| Method | M=20 | M=100 | M=1000 |
| BCD | 0.163±0.028 | 0.241±0.014 | 0.523±0.011 |
| uGLAD | 0.206±0.035 | 0.272±0.024 | 0.569±0.048 |
| AUC | | | |
| Method | M=20 | M=100 | M=1000 |
| BCD | 0.670±0.013 | 0.718±0.014 | 0.839±0.006 |
| uGLAD | 0.774±0.037 | 0.812±0.049 | 0.909±0.040 |

Table 2: **GRN data: non-Gaussian**. AUPR and AUC on 20 test graphs for $D = 100$ nodes and varying samples $M$. Graphs were sampled from the SERGIO simulator for the Gene Regulatory network recovery task. We can observe that the uGLAD model is more adaptive in non-Gaussian settings. A post-hoc masking operation was done to remove all the edges not containing a transcription factor. This was done for all the methods.

| AUPR | | | |
|---|---|---|---|
| Method | M=10 | M=25 | M=50 |
| BCD-avg | 0.137±0.099 | 0.179±0.027 | 0.241±0.045 |
| uGLAD-multi | 0.186±0.028 | 0.204±0.044 | 0.279±0.027 |
| AUC | | | |
| Method | M=10 | M=25 | M=50 |
| BCD-avg | 0.508±0.024 | 0.538±0.024 | 0.597±0.047 |
| uGLAD-multi | 0.552±0.048 | 0.573±0.047 | 0.626±0.022 |

Table 3: **Multi-task learning**. Average AUPR and AUC over $K = 10$ graphs coming from sparsity$\sim [0.05, 0.2]$. The number of nodes $D = 25$ with varying samples $M = [10, 25, 50]$. The BCD-avg considers each instance of the batch as a separate task and reports the average results over the batch. uGLAD-multi is used to recover the graphs jointly using a single model.

| AUPR | | | |
|---|---|---|---|
| Method | dp=0.25 | dp=0.50 | dp=0.75 |
| BCD-mean | 0.583±0.082 | 0.335±0.012 | 0.100±0.009 |
| uGLAD-mean | 0.605±0.103 | 0.357±0.034 | 0.113±0.016 |
| uGLAD-missing | 0.612±0.100 | 0.375±0.043 | 0.132±0.007 |
| AUC | | | |
| Method | dp=0.25 | dp=0.50 | dp=0.75 |
| BCD-mean | 0.792±0.045 | 0.649±0.005 | 0.508±0.009 |
| uGLAD-mean | 0.806±0.019 | 0.691±0.025 | 0.527±0.011 |
| uGLAD-missing | 0.815±0.010 | 0.718±0.002 | 0.560±0.0.41 |

Table 4: **Missing data: Gaussian**. AUPR and AUC on 20 test graphs for $D = 25$ nodes and samples $M = 500$. Gaussian random graphs were generated as described in Sec. 6.1. Increasing fraction of dropouts were introduced to observe the robustness of handling missing data. We can observe that the uGLAD-missing model is more robust, especially in high dropout settings.

| AUPR | | | |
|---|---|---|---|
| Method | dp=0.50 | dp=0.75 | dp=0.90 |
| BCD-mean | 0.468±0.015 | 0.323±0.008 | 0.042±0.017 |
| uGLAD-mean | 0.503±0.011 | 0.346±0.021 | 0.090±0.069 |
| uGLAD-missing | 0.523±0.004 | 0.361±0.043 | 0.117±0.093 |
| AUC | | | |
| Method | dp=0.50 | dp=0.75 | dp=0.90 |
| BCD-mean | 0.819±0.005 | 0.794±0.042 | 0.510±0.010 |
| uGLAD-mean | 0.897±0.009 | 0.821±0.019 | 0.598±0.079 |
| uGLAD-missing | 0.906±0.007 | 0.876±0.013 | 0.706±0.206 |

Table 5: **Missing data: GRN**. AUPR and AUC on 20 test graphs for $D = 100$ nodes and samples $M = 1000$. Gene regulatory network data was used as described in Sec. 6.2. Increasing fraction of dropouts were introduced to the observed samples of the microarray expression data. uGLAD-missing model is more robust for high dropout settings. Such high dropout ratios are quite common in collecting samples for microarray gene expression data.

are hopeful that it improves results over state-of-the-art methods. **Appendix B** applies `uGLAD` to analyse the actual data collected for **anaerobic digestion** and **Appendix C** to analyse data in the **infant mortality** domain.

We use AUC (area under the ROC curve) and AUPR (area under the precision-recall curve) as primary evaluation metrics. The sparsity of the graph leads to very few positive edges. These two metrics account for such imbalance in the data. In addition, they have the advantage of working without specifically setting a threshold for non-zero entries. Their values reported in this work have the mean and the associated standard deviation values listed.

We primarily compare against the BCD algorithm. It has been shown that the other traditional graphical lasso methods like ADMM and G-ISTA gave similar performance as BCD Shrivastava et al. (2020b). We are not aware of any unsupervised deep learning approaches that optimize for the graphical lasso objective.

## 6.1 PERFORMANCE ON SYNTHETICALLY GENERATED GAUSSIAN SAMPLES

The synthetic data was generated based on the procedure similar to the one described in Guillot et al. (2012). A $d$-dimensional precision matrix $\Theta$ was generated by initializing a $d \times d$ matrix with its off-diagonal entries sampled i.i.d. from a uniform distribution $\Theta_{ij} \sim \mathcal{U}(-1, 1)$. These entries were then set to zero based on the sparsity pattern of the corresponding Erdos-Renyi random graph with a certain probability $p$. Finally, an appropriate multiple of the identity matrix was added to the current matrix, so that the resulting matrix had the smallest eigenvalue of 1. In this way, $\Theta$ was ensured to be a well-conditioned, sparse and positive definite matrix and was used in the multivariate Gaussian distribution $\mathcal{N}(0, \Theta^{-1})$, to obtain $M$ i.i.d samples. Table 1 shows the results on this synthetic data.

## 6.2 RECOVERY OF GENE REGULATORY NETWORKS

We conducted an exploratory study to gauge the generalization ability of `uGLAD` to non-Gaussian distributions. We chose the GRN inference task for this purpose. To this end, we use the SERGIO simulator Dibaeinia & Sinha (2020) that provides a list of parameters to simulate cells from different types of biological processes and gene-expression levels with various amounts of intrinsic and technical noise. For evaluation purposes in this work, we created random graphs (GRNs) that were used as input to SERGIO and will act as ground truth for evaluation. First, we set the number of Transcription Factors (TFs) or master regulators. Then, we randomly added edges between the TFs and the other genes based on sparsity requirements. Also, we randomly added some edges between the TFs themselves but excluded any self-regulation edges and maintained connectivity of the graph. When simulating data with no technical noise (clean data), we set the following parameters: sampling-state= 15 (determines the number of steps of simulations for each steady state); noise-param $\sim U[0.1, 0.3]$ (controls the amount of intrinsic noise); noise-type = 'dpd' (the type of intrinsic noise is dual production decay noise, which is the most complex out of all types provided); we set genes decay parameter to 1. The parameters required to decide the master regulators' basal production cell rate for all cell types: low expression range of production cell rate $\sim U[0.2, 0.5]$ and high expression range of cell rate $\sim U[0.7, 1]$. We chose K$\sim U[1, 5]$, where K denotes the maximum interaction strength between master regulators and target genes. Positive strength values indicate activating interactions and negative strength values indicate repressive interactions and signs are randomly assigned. The parameters were configured such that the statistical properties of these synthetic data (clean) set are comparable with the mouse brain Zeisel et al. (2015). Table 2 shows the comparison of the different graph recovery methods on the simulated data generated by SERGIO for cell types $C = 5$ and clean data setting.

## 6.3 MULTI-TASK LEARNING

This experiment verifies the ability of `uGLAD` to do multi-task learning. We chose a collection of tasks as a set of data coming from graphs with varying sparsity. For $K$ different tasks, our input data is $X \in \mathbb{R}^{K \times M \times D}$. We run the `uGLAD` model in multi-task learning mode as described in Sec. 4 and recover $K$ different precision matrices $\Theta \in \mathbb{R}^{K \times D \times D}$ that are optimized for the loss function $\mathcal{L}_{\text{uGLAD-multitask}}(X_K)$ given by equation 6. Table 3 shows results from a single `uGLAD` model in multi-

task setting which is run on the synthetic data generated as described in Sec. 6.1. uGLAD recovers multiple graphs with varying sparsity and shows promise for multitask learning problems.

## 6.4 ROBUSTNESS TESTING FOR MISSING DATA

We artificially introduced missing values or 'dropouts' in the input data $X \in \mathbb{R}^{M \times D}$ to create noisy data. Our aim is to study the effectiveness of the 'consensus' strategy discussed in Sec. 5. We compare it with the baseline statistical imputation technique that does column-wise (or feature-wise) mean imputation as a preprocessing step. BCD-mean and uGLAD-mean, report the results of running the corresponding methods on the column mean imputed data while the uGLAD-missing uses the 'consensus' strategy. Tables 4&5 show the robustness of the 'consensus' strategy introduced in this work for synthetic Gaussian data as well as for the Gene regulatory networks recovery tasks.

## 7  SOFTWARE DETAILS: OPTIMIZING MODES OF uGLAD

To optimize uGLAD for the uGLAD loss function, we have introduced 4 different modes of training. In the software package, these modes can be switched from one to the other using an indicator flag.

*Direct mode*: The input to the uGLAD model is complete data $X$ and the output precision matrix $\Theta = f_{nn}(X)$ is optimized to reduce the uGLAD loss $\mathcal{L}_{\text{uGLAD}}(X)$, as defined in Eq. 4.

*CV mode (recommended)*: In the k-fold cross validation mode, we split the input samples $X = (X_{train}, X_{valid})$. We use the $X_{train}$ as input and optimize for the uGLAD loss $\mathcal{L}_{\text{uGLAD}}(X_{\text{train}})$. Then, we select the best model that minimizes the $\mathcal{L}_{\text{uGLAD}}(X_{\text{valid}})$ for the $X_{valid}$ samples.

*Missing data mode*: We give the entire data $X$ as input with a 'NaN' indicator for the entries where the values are missing. The software then follows the 'consensus' strategy for handling of missing data given in Sec. 5 and outputs the final precision matrix $\Theta^f$.

*Multi-task mode*: Given a batch of input data $X \in \mathbb{R}^{K \times M \times D}$, we jointly optimize them for the uGLAD loss objective to obtain $K$ different precision matrices $\Theta_K \in \mathbb{R}^{K \times D \times D}$, refer to Sec. 4.

## 8  CONCLUSIONS

We introduced a novel technique uGLAD to perform sparse graph recovery by optimizing a deep unrolled network for the graphical lasso objective. This is an extension to the previous GLAD model that was designed to use supervision. The key advantages of using our model over the state-of-the-art algorithms for the graphical lasso problems are (1) Sparsity related hyperparameters are modeled using neural networks which are automatically learned during the optimization. We thus address the sensitivity issue of choosing the right sparsity parameters which is usually a tedious task and often manually set for the other algorithms. (2) By design, neural networks of uGLAD enable the sparsity regularization to be adaptive over the iterations of the optimization leading to superior performance. (3) Our software implementation supports GPU based acceleration and thus can be scaled efficiently to meet the runtime requirements. (4) The uGLAD framework can do efficient multi-task learning. The proposed 'consensus' strategy based on leveraging this property works well to robustly handle missing data. As our experience with anaerobic digestion (see Appendix B) demonstrates, uGLAD can be successfully used as a tool for generating insight into growth dynamics of organisms in a digester and (hopefully) into domain structure in many other applications. We hope that our model becomes one the widely used algorithms to solve the graphical lasso objective.

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
