# OpenReview forum: "uGLAD: A deep learning model to recover conditional independence graphs"
_ICLR.cc/2023/Conference — Submitted to ICLR 2023_

### Official Review · Reviewer_32py · 2022-10-27

**Confidence:** 3
**Correctness:** 4
**Technical Novelty And Significance:** 2
**Empirical Novelty And Significance:** 2
**Recommendation:** 5

**Clarity, Quality, Novelty And Reproducibility:**

The paper is easy to follow, but the contributions of the paper are not significant.

**Strength And Weaknesses:**

Strength:

The authors extended the state-of-the-art GLAD method to the unsupervised setting. Numerical results demonstrated that the proposed method can robustly handle missing data.

Weakness:

1. The extension from GLAD to uGLAD is simple without the need to change the architecture. Therefore, the contributions of this paper are not significant.

2. There are a large number of methods proposed for learning graph models in the literature. More compared methods are needed in experiments.

**Summary Of The Paper:**

This paper considered the problem of learning graph models in an unsupervised setting. The authors extended the state-of-the-art GLAD method to the unsupervised setting, and show its effectiveness in handling missing data by numerical experiments.


**Summary Of The Review:**

The authors extended the state-of-the-art method GLAD to the unsupervised setting. However, such an extension is very simple to conduct, and it is not clear about its significance. The contributions and novelty of this paper are not significant, and experimental comparisons are not sufficient.

---

### Official Review · Reviewer_HUi2 · 2022-10-27

**Confidence:** 4
**Correctness:** 2
**Technical Novelty And Significance:** 3
**Empirical Novelty And Significance:** 2
**Recommendation:** 3

**Clarity, Quality, Novelty And Reproducibility:**

Clarity: The paper's logical flow is fluent, and the method is easy to understand.

Quality: The experiment and result sections lack comparison with state-of-the-art methods and cannot support the paper's claims.

Novelty: This work is an extension of the GLAD model. The methodology exhibits novelty but is not supported well by the results.

Reproducibility: The paper requires more details of the method and settings to run it to be more reproducible.

**Strength And Weaknesses:**

Strength:
+ This work extends the GLAD model to unsupervised and multi-task settings. The adaptive parameter for each precision matrix element provides more thorough controls, and the inductive-biases-based model eases computational burdens. Overall, the method looks promising in theory, and the reasoning behind the extensions is valid and important for real-world applications

Weakness:

- The paper requires a more extensive and robust experimental setup to support its claims.

- For example, in section (page 8), the authors state that the GLAD paper has shown that “ADMM and G-ISTA gave a similar performance as BCD.” So the paper only compares uGLAD with BCD. But there are more sparse Gaussian graphical model estimators. For example, M-estimators [1] and thresholding [2,3] are other types of methods. Although some of these recently proposed methods induce the same regularization coefficients for all precision matrix elements, comparing uGLAD with them is worthwhile to validate adaptive parameter performs better.

- Moreover, this paper tested uGLAD on multi-task data and data with missing values. However, uGLAD is also only compared with BCD. But there are more recent works for these two tasks. For instance, JGL[4] and FASJEM[5] estimate multi-task graphs. MissGLasso[6] and [7] estimate inverse covariance matrices from data with missing values.

- The choice of the simulation is not well justified, and the experimental settings and the results need better descriptions to connect to the paper's contributions.

- Real-world gene expression datasets now have features >25 (in the order of 100s and 1000s). How does the proposed method scale to the large feature size and more noise?

Minor:

- The paper has presented a paper directly from another work (with citation) - this practice is not common in the field and is often discouraged. It would be useful to modify the figure to highlight the novelty and contributions of this work.

- References should have brackets around them when they are not being used as subjects of the sentence.

References:
[1] Yang, Eunho, Aurélie C. Lozano, and Pradeep K. Ravikumar. "Elementary estimators for graphical models." Advances in neural information processing systems 27 (2014).

[2] Sojoudi, Somayeh. "Equivalence of graphical lasso and thresholding for sparse graphs." The Journal of Machine Learning Research 17.1 (2016): 3943-3963.

[3] Zhang, Richard, Salar Fattahi, and Somayeh Sojoudi. "Large-scale sparse inverse covariance estimation via thresholding and max-det matrix completion." International Conference on Machine Learning. PMLR, 2018.

[4] Danaher, Patrick, Pei Wang, and Daniela M. Witten. "The joint graphical lasso for inverse covariance estimation across multiple classes." Journal of the Royal Statistical Society: Series B (Statistical Methodology) 76.2 (2014): 373-397.

[5] Wang, Beilun, Ji Gao, and Yanjun Qi. "A fast and scalable joint estimator for learning multiple related sparse gaussian graphical models." Artificial Intelligence and Statistics. PMLR, 2017.

[6] Städler, Nicolas, and Peter Bühlmann. "Missing values: sparse inverse covariance estimation and an extension to sparse regression." Statistics and Computing 22.1 (2012): 219-235.

[7] Loh, Po-Ling, and Martin J. Wainwright. "High-dimensional regression with noisy and missing data: Provable guarantees with non-convexity." Advances in neural information processing systems 24 (2011).



**Summary Of The Paper:**

This paper extends previous work (GLAD) and proposes a novel deep model, uGLAD, to recover sparse graphs from Gaussian data. The paper introduces the following extensions over the GLAD framework - (1) it extends the framework to recover graphs in unsupervised learning by replacing the loss function (2) it utilizes adaptive regularization parameters (3) by using uGALD in a multi-task setting, the paper describes a consensus strategy to handle missing data. The experiments in the paper show the uGLAD model outperforms traditional block coordinate descent (BCD) on graph recovery of both Gaussian and non-Gaussian (simulated) data.

**Summary Of The Review:**

Overall, the ideas presented are interesting and useful. The paper, however, requires a more descriptive and robust experimental design and results to contribute significantly to the community.

---

### Official Review · Reviewer_aant · 2022-10-28

**Confidence:** 3
**Correctness:** 3
**Technical Novelty And Significance:** 2
**Empirical Novelty And Significance:** 3
**Recommendation:** 5

**Clarity, Quality, Novelty And Reproducibility:**

clarity: mostly clear

quality: ok

originality: poor as  additions are standard.

**Strength And Weaknesses:**

Strength:
Empirical improvement seems significant.

Weakness:
- novelty: although the paper makes several contribution, they are mostly incremental or straightforward. New additions on unsupervised loss function, multi-task extension, and missing value imputation are standard and do not offer any new insight.
- organization: related to novelty, it is not fully clear what problem authors are tackling after reading the paper. Different sections seem to solve different problems, and they seem to be put together just to fill the content of the paper.
- Experiments are only done via one baseline method. Despite authors' claim on other baselines have similar performance, it was only done with specific datasets. Other baselines' results are important here.

Other:
1. citations are not done correctly.
2. "...we expect the linear convergence property holds..." since you change the loss function, how can you ensure this is still the case?
3.


**Summary Of The Paper:**

This paper proposes uGLAD, an unsupervised version of previously proposed GLAD paper, to learn the graph structure for undirected (Gaussian) graphical models. . Authors also added several functionalities on top, including the capability to handle multi-task and missing values. Empirical results show good performance.

**Summary Of The Review:**

incremental improvement over GLAD.

---

### Decision · Program_Chairs · 2023-01-20

**Decision:**

Reject

**Justification For Why Not Higher Score:**

Reviewers believe the technical contribution is rather incremental, and the problem setting could have been made clearer.

**Justification For Why Not Lower Score:**

The studied problem is classical and important, and the paper reports encouraging empirical results.

**Metareview: Summary, Strengths And Weaknesses:**

This paper proposes an unsupervised version of the previously proposed GLAD work to learn the graph structure for undirected (Gaussian) graphical models. The studied problem is classical and important, and the paper reports encouraging empirical results. However, reviewers believe the technical contribution is rather incremental, and the problem setting could have been made clearer.